# Upregulation of Endothelin-1 May Predict Chemotherapy-Induced Cardiotoxicity in Women with Breast Cancer

**DOI:** 10.3390/jcm11123547

**Published:** 2022-06-20

**Authors:** Krithika Krishnarao, Katelyn A. Bruno, Damian N. Di Florio, Brandy H. Edenfield, Emily R. Whelan, Logan P. Macomb, Molly M. McGuire, Anneliese R. Hill, Jordan C. Ray, Lauren F. Cornell, Winston Tan, Xochiquetzal J. Geiger, Gary R. Salomon, Erika J. Douglass, DeLisa Fairweather, Mohamad H. Yamani

**Affiliations:** 1Department of Cardiovascular Medicine, Mayo Clinic, Jacksonville, FL 32224, USA; bruno.katelyn@mayo.edu (K.A.B.); diflorio.damian@mayo.edu (D.N.D.F.); whelan.emily@mayo.edu (E.R.W.); macomb.logan@mayo.edu (L.P.M.); mcguire.molly@mayo.edu (M.M.M.); hill.anneliese@mayo.edu (A.R.H.); ray.jordan@mayo.edu (J.C.R.); salomon.gary@mayo.edu (G.R.S.); douglass.erika@mayo.edu (E.J.D.); fairweather.delisa@mayo.edu (D.F.); yamani.mohamad@mayo.edu (M.H.Y.); 2Department of Cardiovascular Medicine, Ochsner Health, New Orleans, LA 70121, USA; 3Center for Clinical and Translational Science, Mayo Clinic, Jacksonville, FL 32224, USA; 4Department of Cancer Biology, Mayo Clinic, Jacksonville, FL 32224, USA; edenfield.brandy@mayo.edu; 5Department of Oncology, Mayo Clinic, Jacksonville, FL 32224, USA; cornell.lauren@mayo.edu (L.F.C.); tan.winston@mayo.edu (W.T.); 6Department of Pathology, Mayo Clinic, Jacksonville, FL 32224, USA; geiger.xochiquetzal@mayo.edu; 7Department of Environmental Health Sciences, Johns Hopkins Bloomberg School of Public Health, 615 N. Wolfe Street, Baltimore, MD 21205, USA

**Keywords:** biomarkers, chemotherapy-induced cardiotoxicity, angiotensin II type I receptor, endothelin 1

## Abstract

As survival in breast cancer patients from newer therapies increases, concerns for chemotherapy-induced cardiotoxicity (CIC) have offset some of these benefits, manifesting as a decline in left ventricular ejection fraction (LVEF). Patients receiving anthracycline-based chemotherapy followed by trastuzumab are at risk for CIC. Previous research evaluating whether clinical biomarkers predict cardiotoxicity has been inconsistent. Recently, angiotensin II type 1 receptor (ATR1) and endothelin 1 (ET1) have been shown to play a role in breast tumor growth. We evaluated ATR1 and ET1 expression in breast cancer tissue and its association with CIC. A total of 33 paraffin-embedded breast tissue specimens from women with breast cancer treated with anthracycline-based chemotherapy and trastuzumab were analyzed by immunohistochemistry (IHC) and qRT-PCR. We found that ET1 expression was increased in patients with an LVEF ≤ 50% (*p* = 0.032) with a lower LVEF correlating with higher ET1 expression (r = 0.377, *p* = 0.031). In patients with a change in LVEF of greater than 10%, greater ET1 expression was noted compared to those without a change in LVEF (*p* = 0.017). Increased ET1 expression in breast tumor tissue is associated with reduced LVEF. Future studies need to examine whether ET1 may be a tissue biomarker that helps predict the risk of developing CIC in women with breast cancer.

## 1. Introduction

Breast cancer is the most common malignancy diagnosed in women with an incidence around 16% worldwide [1]. Although the advent of newer therapies has improved survival, the risks of chemotherapy-induced cardiotoxicity (CIC) have offset some of these benefits. Approximately 7–18% of patients develop an asymptomatic decline in left ventricular ejection fraction (LVEF) and 0.5–4% develop symptomatic heart failure following the sequential dosing of anthracycline-based chemotherapy and trastuzumab [1]. Of these patients, only about 42% recover their cardiac function [2]. Attempts to minimize CIC have been made, including limiting the dose and frequency of chemotherapy agents, using liposomal formulations, and using dexrazoxane, an iron-chelating agent, in some cases [2,3]. Despite these measures, CIC still remains prevalent.

The accepted definition of CIC, as defined by the Cardiac Review and Evaluation Committee, is considered a reduction in LVEF ≥5% to <55% with symptoms of heart failure or reduction in LVEF ≥10% to <55% without symptoms [1,4,5,6]. The risk of developing CIC increases in those >65 years of age, in females, those with a history of chest radiation, or with co-morbidities and underlying cardiovascular risk factors such as hypertension, diabetes, obesity, hyperlipidemia, those with prior cardiac disease, liver disease, family history of cardiomyopathy, and prior or concomitant cancer therapies including trastuzumab [5,6,7,8,9]. Acute cardiotoxicity occurring within the period of treatment may be reversible and can include the development of pericarditis, myocarditis, or arrhythmias. Chronic cardiotoxicity, on the other hand, manifests later and is typically more serious [7].

The mechanism behind the pathophysiology of CIC development is not well understood. Serum biomarkers such as troponin, natriuretic peptide and other novel biomarkers such as fatty acid binding protein, glycogen phosphorylase isoenzyme, interleukin-6, and myeloperoxidase have been studied as potential predictors of cardiotoxicity; however, studies examining these biomarkers have been inconclusive [1,2,3,6,10]. Recently, multifactorial mechanisms have been described, including involvement of the endothelin pathway signaling and the renin–angiotensin–aldosterone system [11].

Endothelin-1 (ET1) is a potent vasoconstrictor synthesized from the cleavage of big ET1 from pre-proET1 facilitated by endothelin-converting enzyme [12,13]. ET1 has been implicated in arterial hypertension, coronary artery disease, spontaneous coronary artery dissection, Takotsubo cardiomyopathy, angina, arrhythmias, pulmonary arterial hypertension, and heart failure [14]. Furthermore, ET1 and big ET1 levels have been found to be increased in both acute and chronic heart failure [15]. Angiotensin II has well-known direct adverse effects on cardiomyocytes including hypertrophy and adverse remodeling in heart failure [11]. Angiotensin II type 1 receptor (ATR1) is the primary driver of these adverse mechanisms [11].

In addition to their known cardiovascular effects, ET1 and ATR1 have been shown to play a key role in tumor growth in breast cancer [9,16,17,18,19,20,21,22]. ET1 has been implicated in breast tumor growth by stimulating angiogenesis via vascular endothelial growth factor (VEGF) [23,24]. ATR1 is a subtype of the angiotensin II receptor and is also involved in cell proliferation and thought to play a role in tumor angiogenesis [16,17,18]. Based on the limited literature on ET1 and ATR1 expression in breast cancer, the rate of occurrence of these markers in breast cancer is around 43–84% and 10–20%, respectively [16,17,18,19]. Worse outcomes have been demonstrated in patients with breast cancers that express ET1 or ATR1 [12,17,19,20,21]. Evidence suggests that ET1 may influence tumor invasion and metastases, and it may be associated with more relapses of breast cancer years after diagnosis [17,18,19]. ATR1 can promote tumor cell division and proliferation, and tumors expressing this tend to be more aggressive [16].

Given their known cardiovascular effects and their role in tumor growth, angiogenesis, and poor outcomes in breast cancer, we evaluated the relationship of ET1 and ATR1 with the development of CIC. We hypothesized that ET1 and ATR1 expression would be elevated in women with breast cancer and CIC.

## 2. Materials and Methods

### 2.1. Study Population

This was a single center, retrospective study approved by our institutional IRB for analysis of biospecimens and chart review of subjects. A total of 175 patients from 2014 to 2017 who received anthracycline-based agents and subsequent trastuzumab were screened using eligibility criteria from an independent registry maintained within the Department of Oncology in Mayo Clinic Florida. We defined the development of CIC by echocardiography assessment of LVEF from ≥55% at baseline dropping to ≤50% following chemotherapy. Inclusion criteria consisted of women >18 years of age with surgically excised breast tumor, available tissue stored at Mayo Clinic Florida, availability of a sufficient amount of tissue available for analysis, and accessible echocardiographic assessment of LVEF prior to chemotherapy and following completion. Exclusion criteria included males, an incomplete chemotherapy regimen and/or no administration of trastuzumab, LVEF between 51 and 54%, or a drop in LVEF to ≤50% after initiation of chemotherapy with recovery to ≥55% following completion of chemotherapy, tissue samples of poor quality that could not be used for analysis, prior cardiac disease including atrial fibrillation, sustained ventricular tachycardia or structural abnormalities such as congenital heart disease, myocardial infarction, or valvular heart disease.

After the application of eligibility criteria to the initial 175 patients, 52 were identified (Figure 1). Of the 52 patients, 33 paraffin-embedded breast tissue specimens were analyzed as 19 samples had insufficient tissue for analysis. ATR1 and ET1 protein expression in situ was assessed using immunohistochemistry (IHC), and gene expression was quantified with quantitative reverse transcription polymerase chain reaction (qRT-PCR) on breast tumor tissue excised at the time of surgery (i.e., mastectomy, lumpectomy).

### 2.2. Risk Factor Assessment

Demographics were recorded including age, race, co-morbidities and cardiovascular risk factors as well as hormone receptor status for estrogen and progesterone receptor (Table 1). All patients were HER2 positive and had invasive ductal carcinoma. Specific cardiac risk factors and other co-morbidities that may lead to the development of heart failure were identified. Baseline and post-chemotherapy echocardiographic parameters were evaluated focusing on LVEF.

### 2.3. Immunohistochemistry

IHC was performed on breast cancer samples according to standard methods [25]. Tissue sections were de-paraffinized, blocked, and stained with rabbit polyclonal angiotensin II type 1 receptor and anti-angiotensin II type 1 antibody (ab185293, Abcam, Cambridge, MA, USA, 1:200), mouse monoclonal to endothelin-1 (TR.ET.48.5) and anti-endothelin-1 antibody (Ab2786, Abcam, Cambridge, MA, USA, 1:500) or big endothelin-1 (BS-0202R, Thermo Fisher, Waltham, MA, USA, 1:1500) for 1 h at room temperature followed by a biotinylated secondary antibody for 1 h and incubated with 3,3′-diaminobenzidine (DAB) (dark brown). Tissue sections were counterstained with hematoxylin (blue background). Breast tumor tissue sections (5–15 representative sections) were de-identified to blind the analysis; sections of tumor were manually selected using Aperio ImageScope (v12.4.2.7000), and areas of IHC positive tissue was compared to non-positive areas to determine percentage positive normalized to area. Staining positivity was determined using Aperio ImageScope (v.12.4.2.7000). Negative stained representative slides positivity was subtracted from each sample to remove background staining. Positive control tissues were stained for each antibody (ART1: kidney, ET1: bowel, Big-ET1: stomach). A built-in positive pixel count algorithm (v9) was used to determine positivity of the stain. Positivity was derived from an algorithm using the number of positive pixels stained in selected areas, where Np represented the number of positive pixels for stain in all selected areas, Nn represented the number of negative pixels in all selected areas, and Ntotal represented the total positive and negative pixels within the selected areas. Positivity was defined as Np/Ntotal, where the area of positively stained pixels in a selected area is normalized by the total area selected.

### 2.4. qRT-PCR

qRT-PCR was performed on samples according to standard methods [25,26,27]. Human RNA was isolated from formalin-fixed paraffin-embedded (FFPE) blocks from patients with breast cancer. Human RNA was processed using a PureLink FFPE RNA Isolation Kit (Invitrogen, Carlsbad, CA, USA) according to the manufacturer’s instructions. mRNA was converted to cDNA using a High-Capacity cDNA Reverse Transcriptase Kit (Applied Biosystems), as previously described [25,26,27]. qRT-PCR was performed to assess relative gene expression (RGE) using Assay-on-Demand probe sets (Applied Biosystems, Foster City, CA, USA) for ATR1 (AGTR1, Hs05043708), ET1 (EDN1, Hs00174961), ET1 Receptor A (EDNRA, Hs03988672), and ET1 Receptor B (EDNRB, Hs00240747). Primers and reactions were analyzed using the ABI 7000 Taqman system, as previously reported [25,26,27]. Data are shown as RGE normalized to the housekeeping gene, Pumilio RNA Binding Family Member 1 (PUM1, Hs00472881). Additional RT-PCRs were performed utilizing other known housekeeping genes that are used for breast tumor tissues (Symplekin: SYMPK, Hs00191361; Coiled-Coil Serine Rich Protein 2: CCSER2, Hs00982799; Ankyrin Repeat Domain 17: ANKRD17, Hs00289705) [28]. The quality of RNA from paraffin-embedded tissue samples was not of high enough quality for some samples to be able to be analyzed by qRT-PCR, thus leading to a smaller number of samples tested by this method.

### 2.5. Statistical Methods

Continuous variables were summarized as mean (standard error i.e., IHC/qRT-PCR) or standard deviation (hormone data) or range (minimum and maximum) where appropriate, and categorical variables were reported as frequency (percentage). A Wilcoxon rank sum test was used to compare continuous variables between patients with and without CIC, and Fisher’s exact test was used to compare categorical variables between the groups. All tests were two-sided with *p* < 0.05 considered statistically significant. The analysis was conducted using Prism (IHC/qRT-PCR) and R3.6.2 (hormone data).

## 3. Results

### 3.1. Baseline Demographics

During the study period, 52 patients were found to be eligible based on application of inclusion criteria (Figure 1). Nineteen samples did not have sufficient quantity or quality for tissue analysis, leaving 33 that were available for analysis by IHC and qRT-PCR. A total of 14 patients developed CIC with LVEF ≤ 50%, while the remaining 19 preserved normal function with an LVEF ≥55%. The mean age was 62 years and not statistically different between the two groups (Table 1). The majority of patients were White, but this was not statistically different between the groups (75.8% of total, Table 1). Patients developing CIC were therefore more likely to be White (78.6%) followed by Middle Eastern (14.3%) and African American (7.1%) (Table 1).

### 3.2. Hormone Receptor Status

Estrogen receptor (ER) and progesterone receptor (PR)-positive breast tumors were most common within our study population overall (*n* = 16/33, 48.5%,) (Table 1). Women with CIC with an LVEF < 50% were more likely to be ER positive and PR positive (*n* = 8/14, 57.1%) and least likely to be ER positive and PR negative (*n* = 1/14, 7.1%) (Table 1). Compared to those with preserved LVEF, there was no significant difference in women with CIC based on hormone receptor status (*p* = 0.070).

### 3.3. Cardiovascular Risk Factors and Co-Morbidities

Hyperlipidemia was more common in patients with preserved LV function (52.6% vs. 14.3% in those with CIC) in addition to baseline diastolic dysfunction (31.6% vs. 14.3% in those without CIC) (Table 1). Diastolic dysfunction was derived from echocardiographic parameters reported ranging from grade I to IV. All patients with baseline diastolic dysfunction were noted to have grade I dysfunction. Out of five patients with baseline cardiac rhythm abnormalities, defined as left or right bundle branch block (LBBB/RBBB), supraventricular tachycardia (SVT) other than atrial fibrillation, and non-sustained ventricular tachycardia (NSVT), three developed CIC (21.4%, Table 1). Two of these patients had baseline LBBB, and one had NSVT.

### 3.4. Echocardiographic Parameters

Baseline LVEF was similar between the two groups (63% in patients without CIC and 59% in those who developed CIC, Table 1). The lowest mean post-chemotherapy LVEF in patients without CIC was 61% compared to 42% in patients with CIC. Greater change in LVEF from baseline was seen in the CIC group. Pre-chemotherapy global longitudinal strain was not significantly different between the two groups on baseline echocardiography (−21% for patients without CIC compared to −19% in patients with CIC; reference range: more negative than −21%, Table 1). One patient who developed cardiotoxicity had a pre-chemotherapy strain value of −10% with a normal baseline LVEF. Post-chemotherapy strain was −20% on average in the preserved LV function group compared to −16% on average in the CIC group (Table 1). The lowest post-chemotherapy strain in the group without CIC was −17% compared to −11% in the CIC group.

### 3.5. ET1 and ATR1 Expression

Expression of ET1, Big ET1 and ATR1 was assessed by IHC staining of breast tumor tissue (Figure 2). ET1 staining was observed in the cytoplasm of tumor cells with weak to moderate intensity (Figure 2A). ATR1 and Big ET1 staining was found at a low level in tumor cells (Figure 2B,C). Staining positivity was determined using Aperio ImageScope (v.12.4.2.7000). Approximately 30% of tumor tissue sample tested positive for ET1 by IHC compared to the entire tissue sample, while Big ET1 and ATR1 staining was detectable at 10% (Figure 2D). Positive control tissue staining was present at 42.0% in bowel for ET1, 55.2% in stomach for Big ET1 and 16.5% in kidney for ATR1.

By qRT-PCR, ET1 expression was significantly increased in cancer patients with an LVEF ≤ 50% compared to those with an LVEF ≥55% (Figure 3A, *p* = 0.031). Importantly, elevated ET1 expression correlated with a lower percentage of LVEF (Figure 3B, *p* = 0.032). Patients with a reduction in LVEF of 10% or more after the administration of chemotherapy had more ET1 expression compared to those without a reduction (Figure 4A, *p* = 0.026). The greater the reduction in LVEF, the greater the ET1 expression (Figure 4B, *p* = 0.017). There was no significant relationship found between ET Receptor A with LVEF (Figure 5) or change in LVEF (Figure 6), between ET Receptor B with LVEF (Figure 7) or change in LVEF (Figure 8) or between ATR1 and LVEF (Figure 9) or change in LVEF (Figure 10). We also examined whether determining expression levels by qRT-PCR were affected by the type of housekeeping gene that was used (PUM1, SYMPK, CCSER2, ANKRD17). RGE values were found to differ according to the type of housekeeping gene that was used for each gene of interest (i.e., ET1, ET1 Receptor A, ET1 Receptor B and ART1) (Table 2). However, we found that only ET1 was consistently significantly increased compared to the housekeeping gene in breast tissue of patients with CIC according to reduced percentage of LVEF (Table 2).

## 4. Discussion

### 4.1. Patient Population

We did not observe differences in traditional cardiovascular risk factors and co-morbidities between patients with CIC vs. those without in our study population. Interestingly, hyperlipidemia was more prevalent in the preserved LVEF group (*p* = 0.024). Women who developed CIC were more likely to be White; however, this was primarily driven by our patient population where 78.6% (*n* = 25) of our study group was White. One African American patient, out of a total of four, developed CIC in our study. In the two patients of Middle Eastern descent, both developed CIC. Only one patient of Asian descent and one patient of Mexican descent were included in our study population, and neither developed CIC. Larger studies evaluating inherent risks of CIC in relation to ET1 expression based on ethnicity will need to be conducted. Studies such as these may play a role in more tailored therapy for women with breast cancer in this era of individualized medicine where health care disparities remain prevalent.

### 4.2. ET1 Expression

Evidence suggests that ATR1 and ET1 expression occurs specifically in breast tumor tissue of patients with breast cancer [12,16,17,18,19,20,21]. This led to our hypothesis that elevated ATR1 or ET1 expression in breast tissue may be associated with CIC in patients with breast cancer. In our study, 42% (*n* = 14) of women with breast cancer who underwent standard therapy (anthracycline followed by trastuzumab) developed CIC, defined as a drop from preserved baseline LVEF by echocardiography (LVEF ≥ 55%) to LVEF ≤ 50% following chemotherapy. We showed that a greater level of ET1 expression in tumor tissue was associated with CIC utilizing IHC and qRT-PCR. Women with an LVEF ≤ 50% following chemotherapy exhibited higher ET1 expression by qRT-PCR compared to those with a preserved LVEF of ≥55%. Additionally, greater ET1 expression was demonstrated in women with a larger reduction in baseline LVEF of ≥10% following chemotherapy.

These findings suggest the use of ET1 gene expression may help predict those at risk for the development of CIC in patients with breast cancer, but further research is needed. Assessment of this biomarker can be easily performed during the initial biopsy or at the time of surgical resection. Our findings suggest ET1 expression may correlate with CIC as measured by reduced LVEF. These findings also imply that greater ET1 expression may indicate a more severe reduction in LVEF, which is a known poor prognostic marker in heart failure. The mechanism of how ET1 contributes to heart failure remains unclear, but ET1 may be an early biomarker to identify CIC or predict those at risk for developing CIC. Yamashita and colleagues showed that plasma ET1 levels increased in five out of 30 patients during doxorubicin treatment [29]. Two of these five patients developed clinically overt heart failure [29]. Detecting the presence of ET1 in breast tumor tissue at the time of pathologic diagnosis may allow for earlier preventative therapies and closer monitoring for the development of CIC.

Both the endothelin signaling pathway and the renin–angiotensin–aldosterone system are known to exert adverse effects on the cardiovascular system [11,30,31]. These effects are primarily driven by vasoconstriction, volume retention, and impaired myocardial contractility [11,32]. The effect of ET1 as a potent vasoconstrictor is primarily driven by the activation of ETA and ETB receptors [33]. In addition, ETA receptor activation has been associated with cell growth potentiation [33]. More recently, Maayah and colleagues examined ET-1 effects on cardiomyocyte hypertrophy in breast cancer patients [34]. Their findings demonstrated relative cardiomyocyte hypertrophy occurred even prior to systemic breast cancer therapy, suggesting direct effects from breast tumor cells on cardiomyocytes. Although these effects may be independent of those seen in breast cancer patients, its role in the development of CIC needs further evaluation.

In the last three decades, studies have suggested that plasma ET1 levels may be a prognostic biomarker for heart failure [14,35,36,37]. While this has not gained adaptation by consensus guidelines, it is worth further evaluation, particularly in the cardio-oncology population where detailed guidelines are not yet in place. Our study supports following ET1 expression in breast biopsies as a potential method to help predict those at risk for developing CIC. Based on our findings, ET1 expression determined at the time of pathological diagnosis (i.e., lumpectomy, mastectomy) may allow closer follow-up or alteration in therapy to prevent CIC or identify those at risk earlier.

The use of endothelin receptor antagonists is well established in treatment of pre-capillary pulmonary arterial hypertension [38]. However, the use of endothelin receptor antagonists has not been shown to be beneficial in the heart failure population, with some studies demonstrating worse outcomes when used in patients with left heart failure [39,40]. Studies are ongoing evaluating other components of the endothelin signaling pathway as potential novel therapies in cardiovascular disease [41,42]. Recently, Leary and colleagues revealed that elevated ET-1 levels may be associated with a more favorable cardiovascular outcome, suggesting that the role of ET-1 is still not well defined [43]. Targeting the renin–angiotensin–aldosterone system, on the other hand, has been well studied in the heart failure population with therapies focusing on neurohormonal blockade as the standard of care for these patients [31]. Some benefits exist in the use of ACE inhibitors and angiotensin receptor blockers in patients undergoing anthracycline-based chemotherapy; however, these results have not been consistent [44].

### 4.3. Outcomes in Patients Expressing ET1 or ATR1

Elevated ATR1 expression in the breast tissue of CIC patients was not found in our study population. From prior work, patients with HER2 overexpression did not exhibit ATR1 [16,45]. This may explain why no significantly elevated ATR1 staining was seen in our population where all patients expressed HER2. In previous studies, ATR1 also seemed to be associated with ER positive tumors, but breast tumors either overexpressed ATR1 or HER2, but not both simultaneously [16,45]. Previous work has shown anywhere from 10–20% of patients with breast cancer express ATR1 [16]. Several studies have looked into angiotensin receptor blockade in patients receiving chemotherapy to determine if a beneficial response exists but have only shown mixed results [46,47,48,49]. Poor outcomes have been demonstrated in patients with breast cancer who expressed ET1 or ATR1 [12,17,19,20,21]. There is evidence to suggest ET1 may influence tumor invasion and metastases [19,20]. ET1 expression was also noted to have an association with more relapses of breast cancer years after diagnosis [17,18,19]. ATR1 can promote the division and proliferation of tumor cells, and tumors expressing ATR1 are more aggressive [16]. These results may help elucidate the poorer prognosis seen in patients expressing these biomarkers.

### 4.4. Hormone Receptor Status

A review of studies shows mixed results in terms of hormone receptor status and association with ATR1 or ET1. One study found ATR1 expression was observed in patients with ER-positive and HER2-negative tumors, while another study found no association with hormone receptor status [16,17,18]. Other studies have suggested an association between ER status and ET1 expression [19]. The majority of our patients in both groups had ER-positive tumors. Our study was not designed to evaluate hormone receptor status and the development of cardiotoxicity. We did not see a significant relationship between hormone receptor status and ET1 expression or the development of cardiotoxicity in our study. Many ER and PR-negative tumors were associated with cardiotoxicity (35.7%) but there were still more ER and PR-positive tumors overall in patients who developed cardiotoxicity (57.1%). There were more ER-positive tumors in patients with preserved LV function.

## 5. Study Limitations

Our study consisted of a small sample size after screening. Our population may not be well representative of the general population, particularly given the majority of patients were White. We also limited our study to women in order to avoid confounding factors associated with sex differences that have been implicated in the development of cardiomyopathy. Patients with an LVEF between 51 and 54% were excluded despite normal function, which was defined by clinical guidelines within this range due to the presence of other exclusion criteria (atrial fibrillation, mitral regurgitation).

Confounding variables not accounted for in our study include socioeconomic status, diet, exercise, other co-morbidities including HIV status, the presence of obstructive sleep apnea, chronic obstructive lung disease or interstitial lung disease. In addition, we cannot rule out that myocarditis or idiopathic cardiomyopathy resulted in reduced LVEF. This is not a study focusing on the mechanism of developing CIC, but it may allow for future animal studies in order to better understand the relationship of ET1 to CIC.

## 6. Conclusions

By utilizing IHC and qRT-PCR, we were able to demonstrate that greater ET1 expression in breast tumor tissue correlated with an LVEF ≤50% following chemotherapy for breast cancer. In addition, higher ET1 levels were associated with a greater reduction in LVEF compared to baseline. Our findings may aid in future research using novel biomarkers to potentially predict breast cancer patients at risk for developing CIC. In addition, targeted therapies to ET1 may potentially help prevent the development of CIC but will need further evaluation.

Currently, the treatment of CIC manifesting as heart failure is managed similarly to other heart failure patients. Our findings may assist with modifying treatment options to tailor therapy to those who receive chemotherapy. In addition, as one of the first studies to find an association between ET1 and cardiotoxicity in breast cancer patients, our findings could lead to therapies that target these genes to prevent the development of heart failure following chemotherapy.

Based on our findings, following ET1 expression in breast tumor tissue at the time of pathological diagnosis may aid in predicting those at risk for CIC. This may allow for individualized therapy to help prevent CIC or alter the course of treatment earlier. Further studies involving a larger and more diverse population will need to be conducted to evaluate the implications of our findings in detail.

## 7. Perspectives

### 7.1. Competency in Medical Knowledge

In patients with breast cancer treated with anthracycline-based chemotherapy and trastuzumab, the expression of endothelin-1 in breast tumor tissue may predict chemotherapy-induced cardiotoxicity as greater expression was seen in patients with LVEF ≤50% and with a greater reduction in LVEF ≥10% from baseline.

### 7.2. Translational Outlook

Using tissue biomarkers to predict cardiotoxicity may help pave the way for targeted therapies that help treat or even prevent chemotherapy-induced cardiotoxicity.

## Figures and Tables

**Figure 1 jcm-11-03547-f001:**
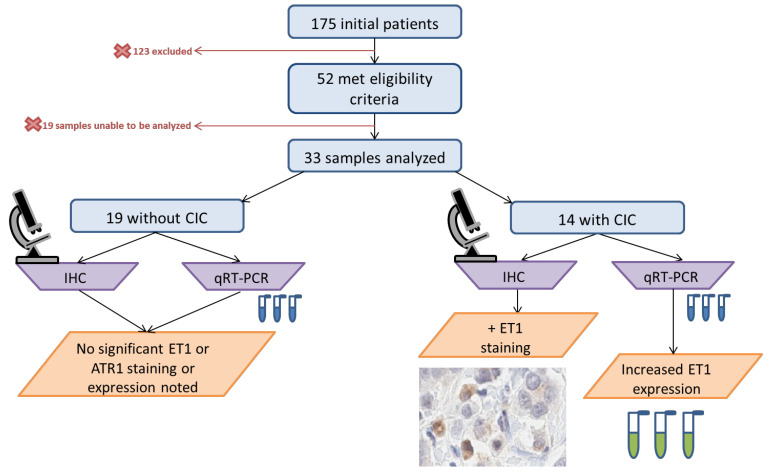
From initial 175 patients that were identified, 52 met eligibility criteria. However, sample that could be used for immunohistochemistry (IHC) and quantitative reverse transcription polymerase chain reaction (qRT-PCR) was only present for 33 samples. Patients with CIC (*n* = 14) were compared to those without CIC (*n* = 19) in analyses.

**Figure 2 jcm-11-03547-f002:**
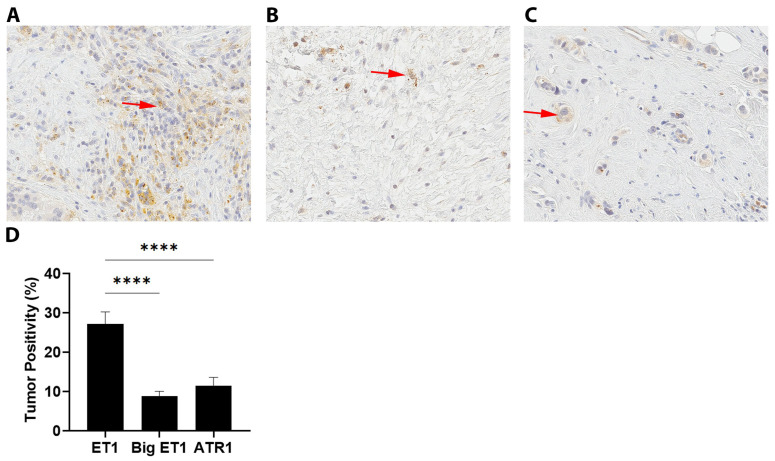
Expression of ET1, Big ET1 and ATR1 staining in breast tumor tissue. (**A**) Representative photo of IHC showing positive staining for ET1 (red arrow). (**B**) Representative photo of IHC showing positive staining for Big ET1 (red arrow). (**C**) Representative photo of IHC staining for ATR1 (red arrow). (**D**) Mean positive immunohistochemistry staining (%) for ET1, Big ET1 and ATR1 in breast tumor tissue normalized to the tumor tissue area. *n* = 33, ****, *p* < 0.0001.

**Figure 3 jcm-11-03547-f003:**
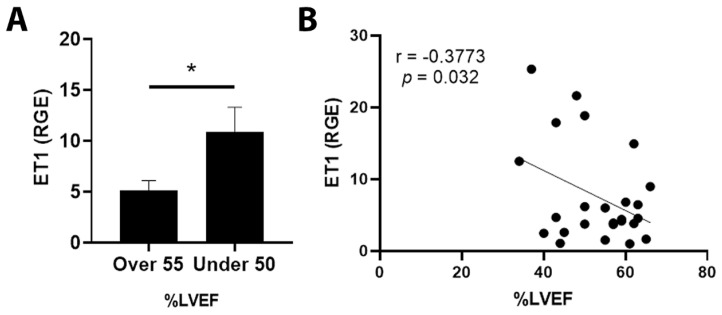
Relationship between LVEF of >55% or ≤50% and ET1 in tumor samples. (**A**) Relative gene expression (RGE) of ET1 by qRT-PCR (* *p* = 0.031) compared to the housekeeping gene PUM1. (**B**) Pearson correlation for ET1 RGE vs. percentage of LVEF showing greater ET1 expression with lower LVEF (* *p* = 0.032). *n* = 25.

**Figure 4 jcm-11-03547-f004:**
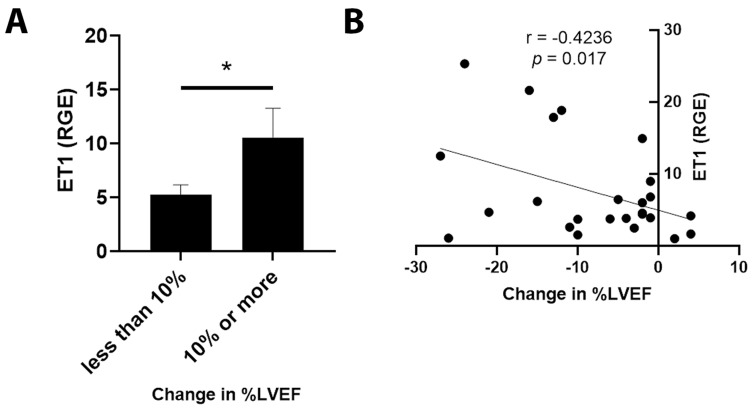
Relationship between change in percentage of LVEF <10% or ≥10% and ET1 expression level. (**A**) Relative gene expression (RGE) of ET1 by qRT-PCR (* *p* = 0.026) compared to the housekeeping gene PUM1. (**B**) Pearson correlation for ET1 expression with change in percentage of LVEF showing more ET1 expression with greater change in LVEF (* *p* = 0.017). *n* = 25.

**Figure 5 jcm-11-03547-f005:**
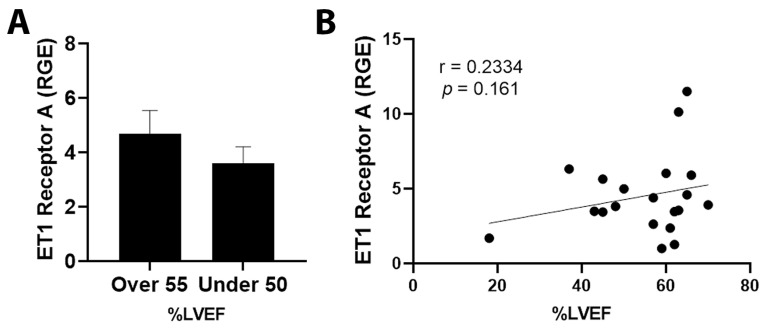
Relationship between LVEF of >55% or ≤50% and ET1 Receptor A in tumor samples. (**A**) Relative gene expression (RGE) of ET1 receptor A by qRT-PCR (*p* = 0.1837) compared to the housekeeping gene PUM1. (**B**) Pearson correlation for ET1 Receptor A vs. percentage of LVEF (*p* = 0.161). *n* = 22.

**Figure 6 jcm-11-03547-f006:**
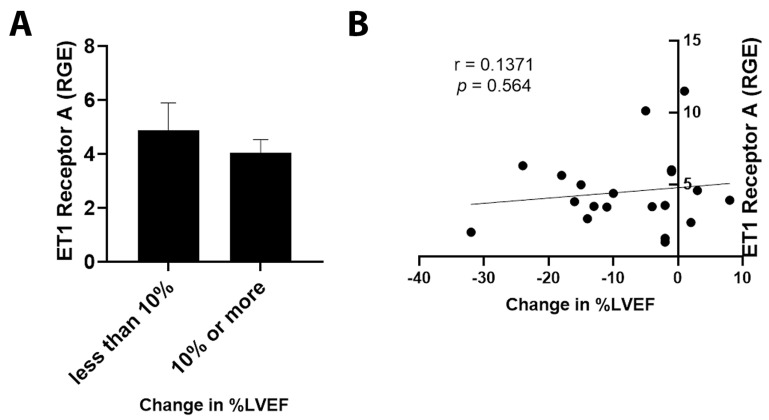
Relationship between change in percentage of LVEF <10% or ≥10% and ET1 Receptor A. (**A**) Relative gene expression (RGE) of ET1 receptor A by qRT-PCR (*p* = 0.248) compared to the housekeeping gene PUM1. (**B**) Pearson correlation for ET1 Receptor A expression vs. change in percentage of LVEF (*p* = 0.564). *n* = 20.

**Figure 7 jcm-11-03547-f007:**
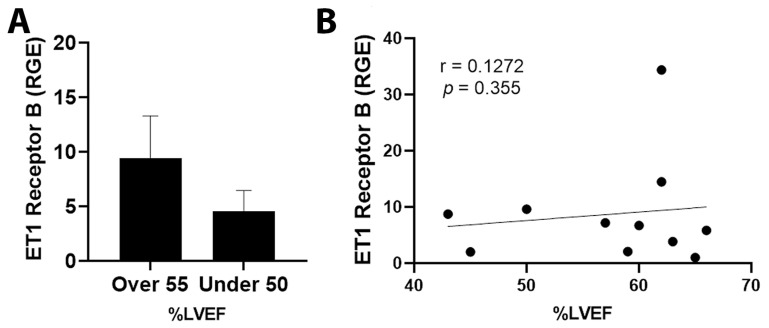
Relationship between LVEF of >55% or ≤50% and ET1 Receptor B in tumor samples. (**A**) Relative gene expression (RGE) of ET1 receptor B by qRT-PCR (*p* = 0.181) compared to the housekeeping gene PUM1. (**B**) Pearson correlation for ET1 Receptor B vs. percentage of LVEF (*p* = 0.354). *n* = 13.

**Figure 8 jcm-11-03547-f008:**
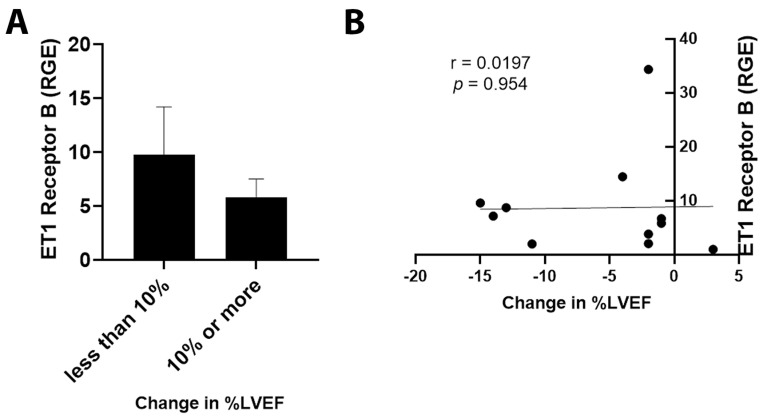
Relationship between change in percentage of LVEF <10% or ≥10% and ET1 Receptor B. (**A**) Relative gene expression (RGE) of ET1 receptor B by qRT-PCR (*p* = 0.242) compared to the housekeeping gene PUM1. (**B**) Pearson correlation for ET1 Receptor B expression vs. change in percentage of LVEF (*p* = 0.954). *n* = 12.

**Figure 9 jcm-11-03547-f009:**
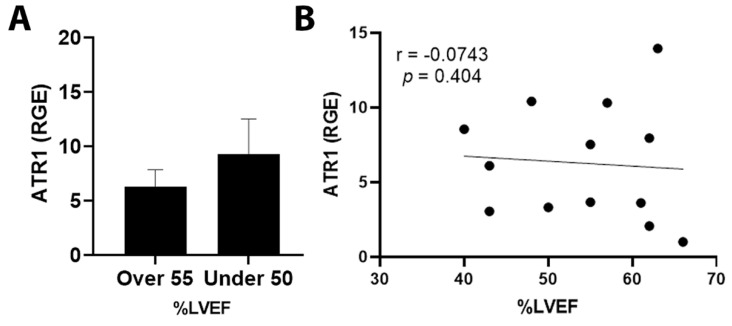
Relationship between LVEF of >55% or ≤50% and ATR1 in tumor samples. (**A**) Relative gene expression (RGE) of ATR1 by qRT-PCR (*p* = 0.379) compared to the housekeeping gene PUM1. (**B**) Pearson correlation for ATR1 vs. percentage of LVEF showing no significant relationship between ATR1 expression and percentage of LVEF (*p* = 0.404). *n* = 13.

**Figure 10 jcm-11-03547-f010:**
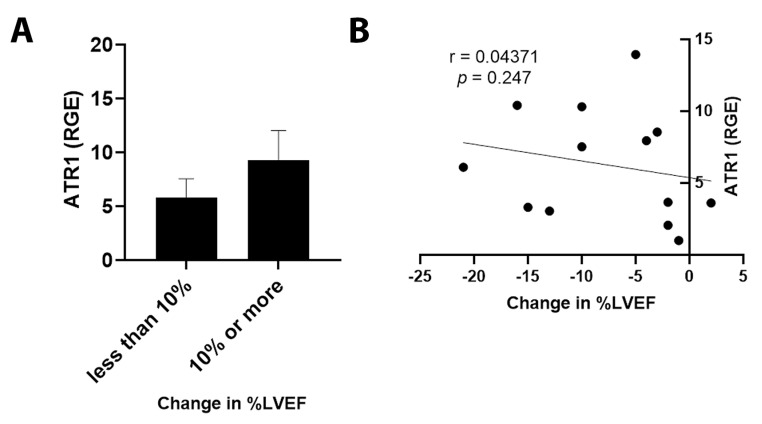
Relationship between change in percentage of LVEF <10% or ≥10% and ATR1. (**A**) Relative gene expression (RGE) of ATR1 by qRT-PCR (*p* = 0.153) compared to the housekeeping gene PUM1. (**B**) Pearson correlation for ATR1 expression vs. change in percentage of LVEF (*p* = 0.247). *n* = 13.

**Table 1 jcm-11-03547-t001:** Baseline characteristics of patients without cardiotoxicity compared to those developing CIC (defined as LVEF ≤ 50% following chemotherapy administration).

	LVEF ≤ 50% (*n* = 14)	LVEF ≥ 55% (*n* = 19)	Total (*n* = 33)	* p * Value
Age				0.098
Mean (SD)	60.0 (10.4)	63.4 (14.1)	62.0 (12.6)	
Median (Range)	59.0 (47.0, 77.0)	62.0 (35.0, 88.0)	62.0 (35.0, 88.0)	
Race				0.318
African American	1 (7.1%)	3 (15.8%)	4 (12.1%)	
Asian	0 (0.0%)	1 (5.3%)	1 (3.0%)	
White	11 (78.6%)	14 (73.7%)	25 (75.8%)	
Mexican	0 (0.0%)	1 (5.3%)	1 (3.0%)	
Middle Eastern	2 (14.3%)	0 (0.0%)	2 (6.1%)	
Coronary artery disease				0.383
No	14 (100.0%)	18 (94.7%)	32 (97.0%)	
Yes	0 (0.0%)	1 (5.3%)	1 (3.0%)	
Hypertension				0.881
No	7 (50.0%)	9 (47.4%)	16 (48.5%)	
Yes	7 (50.0%)	10 (52.6%)	17 (51.5%)	
Hyperlipidemia				0.024
No	12 (85.7%)	9 (47.4%)	21 (63.6%)	
Yes	2 (14.3%)	10 (52.6%)	12 (36.4%)	
Diabetes				0.119
No	14 (100.0%)	16 (84.2%)	30 (90.9%)	
Yes	0 (0.0%)	3 (15.8%)	3 (9.1%)	
Obesity				0.947
No	9 (64.3%)	12 (63.2%)	21 (63.6%)	
Yes	5 (35.7%)	7 (36.8%)	12 (36.4%)	
Baseline LVDD				0.252
No	12 (85.7%)	13 (68.4%)	25 (75.8%)	
Yes	2 (14.3%)	6 (31.6%)	8 (24.2%)	
Conduction abnormalities/arrhythmias				0.226
LBBB	2 (14.3%)	0 (0.0%)	2 (6.1%)	
No	11 (78.6%)	17 (89.5%)	28 (84.8%)	
NSVT, PVCs	1 (7.1%)	0 (0.0%)	1 (3.0%)	
RBBB	0 (0.0%)	1 (5.3%)	1 (3.0%)	
SVT	0 (0.0%)	1 (5.3%)	1 (3.0%)	
Liver disease				0.210
No	14 (100.0%)	17 (89.5%)	31 (93.9%)	
Yes	0 (0.0%)	2 (10.5%)	2 (6.1%)	
Smoking history				0.450
No	7 (50.0%)	12 (63.2%)	19 (57.6%)	
Yes	7 (50.0%)	7 (36.8%)	14 (42.4%)	
Renal disease				0.160
No	11 (78.6%)	18 (94.7%)	29 (87.9%)	
Yes	3 (21.4%)	1 (5.3%)	4 (12.1%)	
Depression/Anxiety				0.518
No	11 (78.6%)	13 (68.4%)	24 (72.7%)	
Yes	3 (21.4%)	6 (31.6%)	9 (27.3%)	
Connective tissue disease				0.823
No	13 (92.9%)	18 (94.7%)	31 (93.9%)	
Yes	1 (7.1%)	1 (5.3%)	2 (6.1%)	
Thyroid disease				0.618
No	12 (85.7%)	15 (78.9%)	27 (81.8%)	
Yes	2 (14.3%)	4 (21.1%)	6 (18.2%)	
Peri-partum period				0.383
No	14 (100.0%)	18 (94.7%)	32 (97.0%)	
Yes	0 (0.0%)	1 (5.3%)	1 (3.0%)	
Aspirin use				0.979
No	11 (78.6%)	15 (78.9%)	26 (78.8%)	
Yes	3 (21.4%)	4 (21.1%)	7 (21.2%)	
FH cardiac disease				0.947
No	9 (64.3%)	12 (63.2%)	21 (63.6%)	
Yes	5 (35.7%)	7 (36.8%)	12 (36.4%)	
ER positive				0.187
No	5 (35.7%)	3 (15.8%)	8 (24.2%)	
Yes	9 (64.3%)	16 (84.2%)	25 (75.8%)	
PR positive				0.393
No	6 (42.9%)	11 (57.9%)	17 (51.5%)	
Yes	8 (57.1%)	8 (42.1%)	16 (48.5%)	
Baseline EF				0.556
Mean (SD)	59% (0.07)	63% (0.04)	61% (0.06)	
Median (Range %)	61% (43, 70)	62% (55, 71)	62% (43, 71)	
Lowest EF				< 0.001
Mean (SD)	42% (0.08)	61% (0.04)	52% (0.11)	
Median (Range %)	43% (18, 50)	62% (55, 70)	57% (18, 70)	
Change in EF				0.007
Mean (SD)	17% (0.08)	1% (0.05)	8% (0.10)	
Median (Range %)	16% (3, 32)	1% (−8, 14)	5% (−8, 32)	
Pre-chemo strain				0.844
Mean (SD)	−19% (0.04)	−21% (0.02)	−20% (0.03)	
Median (Range %)	−20% (−26, −10)	−21% (−25, −18)	−0.20% (−26, −10)	
Most recent strain				0.011
Mean (SD)	−16% (0.04)	−20% (0.03)	−0.18% (0.04)	
Median (Range %)	−16% (−24, −11)	−19% (−26, −16)	−18% (−26, −11)	
ER PR status				0.070
ER-/PR-	5 (35.7%)	3 (15.8%)	8 (24.2%)	
ER+/PR-	1 (7.1%)	8 (42.1%)	9 (27.3%)	
ER+/PR+	8 (57.1%)	8 (42.1%)	16 (48.5%)	

Abbreviations: LVEF: left ventricular ejection fraction, *n*: number, SD: standard deviation, LVDD: left ventricular diastolic diameter, LBBB: left bundle branch block, NSVT: nonsustained ventricular tachycardia, PVCs: premature ventricular contractions, RBBB: right bundle branch block, SVT: supraventricular tachycardia, FH: family history, ER: estrogen receptor, PR: progesterone receptor, EF: ejection fraction.

**Table 2 jcm-11-03547-t002:** Relationship between LVEF of >55% or ≤50% and genes of interest using different housekeeping genes in tumor samples.

House Keeping Gene	Gene of Interest	≥55% LVEF (RGE)	<50% LVEF (RGE)	*p*-Value
PUM1				
	ET1	5.2 ± 0.9	10.8 ± 2.4	** *0.03* **
	ET1 Receptor A	7.3 ± 2.8	3.6 ± 0.6	0.10
	ET1 Receptor B	9.4 ± 3.8	4.5 ± 1.9	0.14
	ART1	6.3 ± 1.6	9.3 ± 3.2	0.43
SYMPK				
	ET1	6.5 ± 1.3	11.9 ± 2.3	** *0.04* **
	ET1 Receptor A	2.9 ± 0.3	2.1 ± 0.4	0.05
	ET1 Receptor B	38.6 ± 10.2	25.1 ± 6.3	0.14
	ART1	36.7 ± 13.1	35.3 ± 14.4	0.47
CCSER2				
	ET1	5.5 ± 0.9	9.4 ± 3.9	0.17
	ET1 Receptor A	5.5 ± 1.6	3.4 ± 0.9	0.14
	ET1 Receptor B	10.8 ± 3.1	11.0 ± 6.5	0.48
	ART1	72.5 ± 22.9	27.3 ± 6.7	0.05
ANKRD17				
	ET1	7.6 ± 1.2	14.2 ± 3.3	** *0.04* **
	ET1 Receptor A	9.4 ± 2.7	10.0 ± 2.3	0.43
	ET1 Receptor B	4.6 ± 0.7	4.7 ± 1.7	0.48
	ART1	7.2 ± 1.4	27.1 ± 18.8	0.16

Bold and italic indicate *p* values that are significantly different comparing ≤55% LVEF to >55% LVEF.

## Data Availability

The data presented in this study are available on request from the corresponding author. The data are not publicly available due to containing patient data.

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
