# Peer review of "Upregulation of Endothelin-1 May Predict Chemotherapy-Induced Cardiotoxicity in Women with Breast Cancer"

_jcm, 2022, doi:10.3390/jcm11123547_

Round 1
Reviewer 1 Report
Thanks to the authors for addressing all comments. No more comments.
Author Response
We greatly thank the reviewers for their time and consideration of our research. We appreciate their constructive comments to allow our manuscript to be revised and resubmitted.
Reviewer 2 Report
1. Line 199-200: From table 1, ER-/PR- patients had higher chance to develop CIC (5/8=62.5%) , the incidence is higher than ER+/PR+ group (8/16=50%)
2. Line 217 What does it mean " the lowest absolute LVEF value"? What is the difference between the lowest LVEF and the lowest absolute LVEF ?
Author Response
- Line 199-200: From table 1, ER-/PR- patients had higher chance to develop CIC (5/8=62.5%) , the incidence is higher than ER+/PR+ group (8/16=50%)
- Thank you for reviewing this carefully. We have made changes to line 199-201 to read as follows with Track Changes. The CIC group is the first column in Table 1 that included 14 patients as the denominator.
- “Women with CIC with an LVEF <50% were more likely to be ER positive and PR positive (n=8/14, 57.1%) and least likely to be ER positive and PR negative (n=1/14, 7.1%) (Table 1).”
- Line 217 What does it mean "the lowest absolute LVEF value"? What is the difference between the lowest LVEF and the lowest absolute LVEF ?
- We apologize for this not being clearly defined or reflected within Table 1. The lowest LVEF reported in Table 1 is reported as a mean. The lowest absolute LVEF is actually not reported in Table 1 but represents the lowest LVEF number noted by echocardiography in patients with CIC. Since this is not reported in Table 1, we removed this comment. We also clarified that the lowest LVEF is reported as a mean. Thank you for catching this.
Reviewer 3 Report
In the present manuscript Krishnarao and co-workers have addressed an important and currently relevant health issue such as the cardio-toxicity induced by anticancer drugs. This study is of special interest regarding the search for early markers since cardio-toxicity usually appears years after surviving the cancer. Although this is a retrospective study, with usual limitations, it opens the doors for further trials assessing the potential usefulness as a predictor of cardio-toxicity by early histology in breast cancer patients.
In addition, I have carefully read previous review and up to my view authors has answered satisfactorily previous request. Therefore, I have no extra suggestions to do, rather than congratulate authors for this study.
Author Response
Thank you for taking the time to review and re-review our manuscript in detail. We greatly appreciate your thoughtful comments and suggestions.
This manuscript is a resubmission of an earlier submission. The following is a list of the peer review reports and author responses from that submission.
Round 1
Reviewer 1 Report
In the current study, the authors performed a retrospective analysis of breast cancer samples and investigated whether the expression of ET-1 and ATR-1 can be predictive of cardiotoxicity following cancer treatment. Based on their results they concluded that ET-1 can be an early marker to predict CIC in breast cancer patients. The study design and analysis sound reasonable. However, further improvements can be done to further support the conclusions.
Major comments
- While the expression of ET-1 in initial breast cancer biopsy can be an early marker for CIC as the authors suggested, it would be great if they can investigate if there is a correlation also between the circulating levels of ET-1 in plasma prior to the start of therapy and CIC.
- Big ET-1 was shown previously to be more stable than ET-1. Have the authors considered measuring its expression?
- The authors used the ATR1 antibody by Abcam97571 for IHC which has been discontinued. Have the authors investigated different ATR1 anti-body before concluding that it has no expression in breast cancer tissues?
- The number of samples evaluated by qRT-PCR for ATR1 (n=13) is different from ET1 (n=25). What is the explanation for this discrepancy?
- In the discussion, the authors have provided limited information about the role of ET-1 in the heart and don’t provide an adequate explanation why the expression of ET-1 in breast cancer biopsy can be higher in patients with CIC. Although they mentioned that the study is not mechanistic, adding more evidence about the general effect of tumors on the heart and the specific role of ET-1 is necessary to support their findings and improve the scientific soundness. This recent article has discussed relative hypothesis and should be cited:
- Maayah, Z.H., et al., Breast cancer diagnosis is associated with relative left ventricular hypertrophy and elevated endothelin-1 signaling. BMC Cancer, 2020. 20(1): p. 751.
- The authors suggested that the findings could lead to therapies that target these genes. Please, consider adding more insights if there are current strategies to inhibit ET-1 and whether some of these strategies have been investigated and provided better protection in the context of cardiovascular diseases.
- The authors reported new results in Section 3 of the discussion “Hormone receptor status” that were not reported in the results section and not supported by graphs/tables. So, I suggest rewriting it again. I believe the discussion should refer to results already reported in the results section.
Minor comments:
- Add abbreviations in the footnotes of table 1.
- Line 194: “The lowest post-chemotherapy LVEF in patients without CIC was 63% compared to 42% in patients with CIC.” It should be 61%, not 63% according to table 1.
- Figure 1C: It is better to be consistent with the names of markers in the figure and the caption (ET1 and ATR1 vs EDN1 and AGTR1).
- Line 211: I suggest reporting the p values as it is in figure 2A (p=0.031) not <0.032.
- Figure 3A and 5A: I suggest adding the x-axis title (change in %LVEF) similar to figure 3B and 5B.
- Figure 3A and 5A change in LVEF% reported as 10% or more, while the text the authors used >10%. So, I suggest changing it throughout the paper to match the figure (≥10%).
- Line 264 “ATR1 expression was not found in our study population”. That is a very general sentence. I suggest being specific in the method of detection and the sample type.
Reviewer 2 Report
I have had the pleasure of peer reviewing the article "Upregulation of endothelin-1 may predict chemotherapy-induced cardiotoxicity in women with breast cancer". Congratulations to the authors for the methodological quality, the correct presentation of the results and the clarity of the exposition. This study is a further step in the study of biomarkers that can predict the risk of CIC secondary to chemotherapy in patients with breast cancer .
For this reason, and although it is a study with few patients, with biases associated with its retrospective nature and the results will not change clinical practice or the routine use of ET1 or TKA, given the quality of presentation and clinical relevance.
Reviewer 3 Report
The authors reported on the association between endothelin-1 and angiotensin ii receptor in women with breast cancer. 52 patients who received anthracycline agents and subsequent trastuzumab were tested retrospectively for endothelin-1 and angiotensin ii type 1 receptor expression. Immunohistochemistry and qPCR were performed and association with cardiotoxicity was assessed. The authors found that a higher endothelin-1 expression was positively associated with a decreased left ventricular ejection fraction.
The study is sound but they are issues that should be clarified.
Major remarks:
- The authors studied the correlation between endothelin-1 expression and cardiovascular parameters in breast cancer. Two articles from Wülfing P et al. from 2003 and 2004 showed that not only endothelin-1 but also its receptors, endothelin-1 type A and B, are involved in breast cancer development. As the authors tested angiotensin II type 1 receptor expression, it would be advisable to also test andothelin-1 type A and B receptors expression.
- In the results part about the clinical features of the patients, there are no indications to understand which parameters were statistically significant or not. This part should be revised so that the reader gets what is different between the patient groups.
- Concerning the immunohistochemistry labeling, the authors did not include any positive or negative controls. It would be interesting to put the results in perspective by comparing them with tissues that express or not endothelin-1 and angiotensin II type 1 receptor. Moreover, different articles have reported unspecificity of commercial antibodies targeting angiotensin II type 1 receptor. Have the authors tested different antibodies for this receptor?
- Finally, the authors used the PUM1 gene as housekeeping gene for their qPCR experiments. One recent article showed that PUM1 is a possible marker for breast cancer (Murillo Carrasco A et al., J Clin Lab Anal 2021). Another revealed that a combination of different housekeeping genes including PUM1 is more robust than genes alone (Tilli TM et al., BMC 2016). Have the authors analyzed other housekeeping genes as PUM1? It would be recommendable to evaluate the impact of different housekeeping genes on the results and observe if the use of these genes impact the results.